# Periductal bile acid exposure causes cholangiocyte injury and fibrosis

**Miri Dotan**[1,2], **Sophia Fried**[1], **Adi Har-Zahav**[1], **Raanan Shamir**[1,2], **Rebecca G. Wells**[3], **Orith Waisbourd-Zinman**[1,2]*

**1** Sackler Faculty of Medicine, Tel-Aviv University, Tel Aviv, Israel, **2** Pediatric Gastroenterology, Nutrition and Liver Diseases, Schneider Children's Medical Center, Petach Tiqva, Israel, **3** Departments of Medicine, Pathology and Laboratory Medicine, and Bioengineering, University of Pennsylvania, Philadelphia, Pennsylvania, United States of America

* oritwz@gmail.com

## Abstract

### Introduction

Bile duct integrity is essential for the maintenance of the structure and function of the biliary tree. We previously showed that cholangiocyte injury in a toxic model of biliary atresia leads to increased monolayer permeability. Increased epithelial permeability was also shown in other cholangiopathies. We hypothesized that after initial cholangiocyte injury, leakage of bile acids into the duct submucosa propagates cholangiocyte damage and fibrosis. We thus aimed to determine the impact of bile acid exposure on cholangiocytes and the potential therapeutic effect of a non-toxic bile acid.

### Materials and methods

Extrahepatic bile duct explants were isolated from adult and neonatal BALB/c mice. Explants were cultured with or without glycochenodeoxycholic acid and ursodeoxycholic acid. They were then fixed and stained.

### Results

Explants treated with glycochenodeoxycholic acid demonstrated cholangiocyte injury with monolayer disruption and partial lumen obstruction compared to control ducts. Masson's trichrome stains revealed increased collagen fibers. Myofibroblast marker α-SMA stains were significantly elevated in the periductal region. The addition of ursodeoxycholic acid resulted in decreased cholangiocyte injury and reduced fibrosis.

### Conclusions

Bile acid leakage into the submucosa after initial cholangiocyte injury may serve as a possible mechanism of disease propagation and progressive fibrosis in cholangiopathies.

**Data Availability Statement:** The dataset is available on Kaggle (DOI: 10.34740/kaggle/dsv/3208167).

**Funding:** The study was funded in part by US-Israel BSF [grant #2017-212]; and by Fred and

Suzan Biesecker Pediatric Liver Center grants. The funders had no role in study design, data collection and analysis, decision to publish, or preparation of the manuscript.

**Competing interests:** The authors have declared that no competing interests exist.

**Abbreviations:** BA, biliary atresia; EHBD, Extrahepatic bile duct; GCDCA, glycochenodeoxycholic acid; PBS, phosphate-buffered saline; UDCA, ursodeoxycholic acid.

## Introduction

Bile acids are amphipathic end products of cholesterol metabolism [1]. Bile acid composition varies substantially among animal species. In cholestasis, impaired bile flow leads to the accumulation of bile acids, which causes injury and inflammation [2]. Bile acids have intrinsic membranolytic properties, yet under physiologic conditions, cholangiocytes are protected from bile acid toxicity on their apical side. However, apical and basolateral cholangiocyte plasma membranes differ in their lipid and protein composition and fluidity [3]. We hypothesized that an initial insult causes increased epithelial permeability; this results in a bile leak to the basolateral side, which may be more susceptible to the toxic effects of bile acids. This subsequently propagates cholangiocyte injury and periductal fibrosis [4].

We have previously shown that in a toxic model of biliary atresia (BA) there is increased epithelial permeability [5]. BA is a neonatal liver disease that occurs in 1:5000–1:18,000 live births around the world [6]. Severe extrahepatic bile duct (EHBD) fibrosis is usually present at the time of diagnosis and the extrahepatic cholangiocyte injury is significantly more pronounced at the time of diagnosis compared to intrahepatic cholangiocytes [7].

Previous work led to the identification of a plant toxin, biliatresone, which causes selective EHBD damage in zebrafish and a BA-like disease in the offspring of livestock exposed in pregnancy [8]. Biliatresone treatment of murine cholangiocyte spheroids leads to rapid loss of cellular tubulin, increased epithelial monolayer permeability, loss of apical polarity, and monolayer disruption [4, 5]. Cholangiocytes in a microfluidic bile duct-on-a-chip showed increased monolayer permeability in response to biliatresone treatment, and this was worse when the application of biliatresone was to the basolateral surface [9].

Chronic cholestatic liver diseases, such as primary biliary cholangitis (PBC) and primary sclerosing cholangitis (PSC), are often associated with alterations in the tight junctions of cholangiocytes and biliary epithelial cells [10]. E-cadherin is an important adhesion molecule whose loss in knockout mice was associated with periportal inflammation as well as periductal fibrosis, which resembled primary sclerosing cholangitis [11].

## Materials and methods

### Use of experimental animals

An animal research ethics committee prospectively approved this research (Tel-Aviv University and the Israel Ministry of Health, license number 01-16-098). All the mice that were used were under strict standards of care and experimental planning. The study was administrated in compliance with the ARRIVE guidelines for the involvement of animals in the study. Adult mice were euthanized at six weeks of age or older using carbon dioxide, and neonatal mice were euthanized at the age of 3 days by isoflorane, both followed by cervical dislocation.

### Bile acids and EHBD cultures

The concentration of bile acids varies depending on the location in the enterohepatic circulation. The concentration of bile acids in the canaliculus and biliary ductules is high, at 20–40 mmol/L. In the gallbladder, the concentration increases up to 200 mmol/L. These high concentrations are mandatory for sufficient micelle formation [1, 4]. Previous work found that glycochenodeoxycholic acid (GCDCA) concentration in the bile of cholestatic patients was around 4–5 mM [12, 13].

EHBDs were isolated from adult and 3-day-old mice, employing a protocol that was previously described [14]. The EHBDs were treated with GCDCA 5 or 50mM (G0759 Merck) and

ursodeoxycholic acid (UDCA) at 5mM (U5127 Merck). Control ducts were immersed in a biliary epithelial cell medium.

The pH was adjusted to physiological levels. The ducts were incubated for 24 hours at 37°C, 95% $O_2$ and 5% $CO_2$ in a Vitron Dynamic Organ Culture Incubator (San Jose, United States). All the experiments were repeated a minimum of three times, with two technical replicates each.

## Staining and imaging

Following the incubation, adult ducts were inserted in a solution of 2% bacto agar and 2.5% gelatin at 4°C for 1 hour, and then placed inside histology cassettes in 70% alcohol. The ducts were fixed in paraffin blocks and sectioned for staining (7 µm sections).

Neonatal ducts were fixed using a whole-mount staining technique, as previously described [15]. EHBD were fixed in 4% formalin for 15 minutes and washed with phosphate-buffered saline (PBS). Then they were permeabilized in Dent's fixative (80% methanol/20% dimethyl sulfoxide) for 15 minutes and gradually rehydrated with a series of methylene and water solutions. Later, the ducts were washed with PBS and diluent (1× PBS with 0.1% Triton X-100 and 1% goat serum) for 1.5 hours. This was followed by blocking in a diluent solution containing 10% normal goat serum for an additional 2 hours.

Both neonatal and adult ducts were stained using antibodies against the cholangiocyte marker keratin 19 (K19, 1:10, Developmental Studies Hybridoma Bank, TROMAIII) and for the myofibroblast marker α-smooth muscle actin (mouse α-SMA, Abcam ab7817). Adult bile ducts were also stained for hematoxylin and eosin and Masson trichrome.

Leica SP5 confocal microscope at 40X magnification was used for imaging of neonatal ducts and Axioimager Z2 apotome microscope at 20X-40X for adult ducts. Images were analyzed by using FIJI ImageJ software (https://imagej.net/Fiji/Downloads) employing a standard color threshold. The extent of fibrosis was compared between samples by measuring the thickness of collagen staining surrounding the lumen. In Masson's trichrome-stained bile ducts, collagen deposition was expressed as the percentage of collagen positive area with relation to the whole duct wall. Quantification of total α-SMA positive area in the bile ducts was automated with a custom ImageJ macro (MeasureSignalWidthV5.ijm) with values that were normalized to control.

Relevant regulations and guidelines were used for all methods.

## Results

To determine the effect of bile acids on the surface epithelium of adult cholangiocytes, we added the toxic hydrophobic bile acid GCDCA at 5 mM, to the cholangiocyte media of adult mouse EHBD explants. Ducts treated with GCDCA demonstrated altered duct morphology: cholangiocytes with abnormal nuclear chromatin and disruption of the cholangiocyte monolayer, with visible cell sloughing. This resulted in partial lumen obstruction compared to control ducts, which remained intact (Fig 1a, upper panel). The width of the collagen layer in the submucosa of the GCDCA-treated ducts, as highlighted by Masson's trichrome staining, was significantly increased (Fig 1a, middle panel and Fig 1b). K19 staining, which is specific for cholangiocytes, further demonstrated loss of the epithelial monolayer, with loss of cell-to-cell adhesion and areas of cholangiocyte clustering (Fig 1a, lower panel). Immunofluorescence staining for α-SMA also demonstrated significantly increased staining in the periductal region in the GCDCA-treated ducts (Fig 1a, lower panel and Fig 1c).

Next, we determined the effect of GCDCA on the surface epithelium of neonatal ducts. We previously showed, in a toxic model of BA, increased epithelial permeability [5]. Hence, we

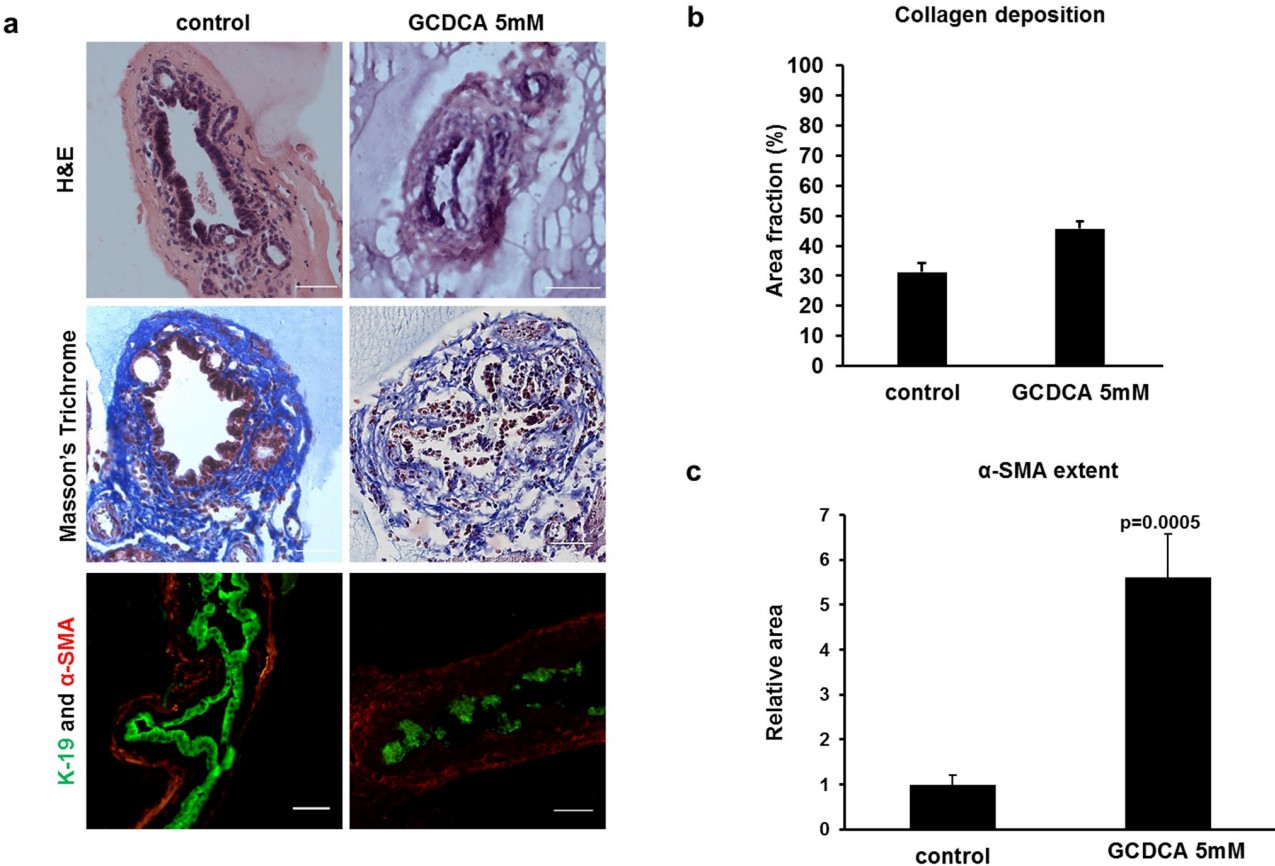

**Fig 1. Glycochenodexycholic acid (GCDCA) causes cholangiocyte injury and subepithelial fibrosis in mice extrahepatic bile ducts (EHBDs).** (a) EHBDs were dissected and incubated for 24 hours with and without GCDCA at 5mM. The ducts were then sectioned and stained for hematoxylin and eosin (upper panel), Masson's trichrome (middle panel), and immunofluorescence: the cholangiocyte marker K19 (green) and the myofibroblast marker α-SMA (red) (lower panel). Disruption of the cholangiocyte layer was observed with all three staining modalities. Marked fibrosis was evident, with a thickened collagen layer highlighted with Masson's trichrome, and with increased immunofluorescent stain with α-SMA. Scale bar, 50 μm. (b) The extent of collagen deposition was expressed as the proportion (%) of Masson's trichrome stained area with respect to the total biopsy area (control 31.2% ± 2.93, GCDCA 5mM 44.53% ± 2.23, GCDCA 50mM 71.8% ± 5.13, (n = 18 ducts)). (c) Quantification of the total relative α-SMA positive area in the bile ducts (control 1 ± 0.213 (n = 24), GCDCA 5.6104 ± 0.972, (n = 16)). Data represent mean ± standard error of the mean, N = 4–6 independent experiments.

were interested in observing the response to bile acid leakage in the EHBD of neonatal mice. We treated EHBDs of three-day-old mice with GCDCA. Since neonatal EHBDs can be stained whole mount, we were able to use confocal microscopy to examine changes throughout the duct and to assess lumen integrity. This technique is not feasible with larger ducts. GCDCA caused extensive lumen obstruction and periductular fibrosis in neonatal EHBDs compared to untreated neonatal EHBDs, which remained intact (Fig 2). Additionally, GCDCA-treated ducts had notably greater α-SMA staining in the periductal region (Fig 2a).

We were also interested in assessing whether the addition of UDCA to GCDCA alters the effects of GCDCA on the neonatal EHBD. Neonatal EHBD treated with both GCDCA and UDCA demonstrated decreased cholangiocyte injury, improved lumen integrity and significantly decreased fibrosis compared to treatment with GCDCA alone, as reflected by reduced α-SMA expression (Fig 2a). This suggests that UDCA can attenuate bile acid toxicity in diseases with increased epithelial permeability.

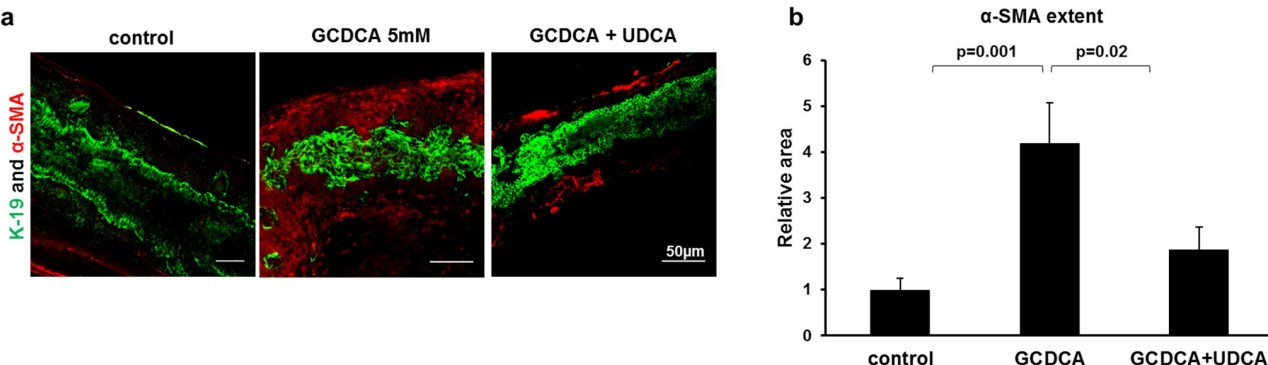

**Fig 2. Glycochenodexycholic (GCDCA) causes lumen obstruction and subepithelial fibrosis of neonatal extrahepatic bile ducts (EHBDs), while ursodeoxycholic acid (UDCA) attenuates GCDCA toxicity.** (a) Neonatal EHBD were dissected and incubated for 24 hours in biliary epithelial cell media, with or without GCDCA 5mM, and with GCDCA 5mM combined with UDCA acid 5mM. Immunofluorescent staining for the cholangiocyte marker K19 (green) and the myofibroblast marker α-SMA (red) demonstrated an ameliorating effect of UDCA, with increased lumen integrity and decreased fibrosis. Scale bar, 50 μm. (b) Quantification of the total α-SMA positive area in the bile ducts (control 1 ± 0.255 (n = 20), GCDCA 4.19 ± 0.87 (n = 21), GCDCA +UDCA 1.87 ± 0.48 (n = 15)). Data represent mean ± standard error of the mean, N = 4 independent experiments.

## Discussion

Bile duct integrity is essential for the maintenance of the structure and function of the biliary tree. Disruption of tight junction integrity is part of the pathogenesis of biliary diseases such as ischemic cholangitis, primary biliary cholangitis, primary sclerosing cholangitis, hepatocellular carcinoma and cholangiocarcinoma [16]. Iatrogenic bile duct injury during surgery, leading to bile leaks, is associated with significant perioperative morbidity and mortality [17].

The apical surface of cholangiocytes confronts and normally resists the hostile luminal environment, which contains millimolar concentrations of bile acids; the basolateral surface is not exposed to bile and its contents. However, an injury may disrupt cholangiocyte tight junctions and epithelial barrier function, leading to bile leakage into the duct submucosa, with exposure of the basolateral surface of cholangiocytes to bile [16]. Here we showed that bile acid exposure results in cholangiocyte injury and fibrosis.

In BA, the role of bile acids in fibrosis progression is yet to be established. BA is a cholangiopathy that rapidly progresses to cirrhosis [6, 18–21]. In a toxic model of BA, we and colleagues previously showed increased epithelial permeability of injured cholangiocytes, with decreased expression and abnormal localization of the apical tight junction protein ZO-1 [5]. It is possible that an initial insult causes increased permeability, leading to the leakage of toxic bile acids. In the second stage, exposure to toxic bile acids of the basolateral side of the duct may lead to propagation of injury.

We used a novel bile duct explant culture system to delineate the effects of bile acids. We showed that GCDCA causes significant cholangiocyte injury, leading to lumen obstruction and fibrosis. GCDCA is a primary conjugated biliary acid and is known to be toxic and to accumulate in cholestasis [22]. GCDCA deregulates autophagy and causes abnormal expression of mitochondrial antigens, followed by cellular senescence in cholangiocytes [23]. Previous studies demonstrated that GCDCA caused apoptosis in rat liver cells and necrotic cell death in human cells [4]. In our model, GCDCA damage was seen in both neonatal and adult mouse EHBD.

We demonstrated that UDCA ameliorates the toxic effects of GCDCA in EHBD explants. UDCA is a secondary bile acid with choleretic properties. It is used therapeutically in cholestatic liver diseases [24–28]. In cholestasis, hydrophobic bile acids damage ductal cell

membranes, while UDCA is a relatively hydrophilic bile acid. Moreover, UDCA exerts anti-inflammatory and protective effects on human epithelial cells of the gastrointestinal tract and has been linked to immunoregulatory responses [29]. Sakisaka et al. described alteration in the tight junction protein 7H6 in livers of patients with primary biliary cirrhosis. While in untreated patients, immunostaining for 7H6 was diminished to absent, in livers of patients treated with UDCA, immunostaining was well preserved [10]. Our findings are consistent with the observed beneficial effects of UDCA in cholestatic patients.

This research has several limitations. First, bile acid composition, metabolism, and toxicity vary markedly between species. Therefore, findings in mice may not be directly extrapolated to humans [4]. We showed the effects of only two bile acids; initial experiments with cholic acid showed milder damage than GCDCA, and further research is needed. Lastly, the EHBD explants were cultured in a system that exposes GCDCA not only to the basolateral domain but also to the apical domain of cholangiocytes. We hope that in the future, novel study systems [9] will enable us to differentiate between the apical and basolateral sides and better demonstrate the effects of bile acids on each domain.

In summary, exposure of EHBDs to GCDCA caused cholangiocyte injury and fibrosis. We suggest that this represents a mechanism of propagation of injury after increased epithelial permeability due to initial cholangiocyte damage. More research is needed to characterize disease progression in humans and to determine whether the injury can be reversed at early stages.

## Author Contributions

**Conceptualization:** Miri Dotan, Raanan Shamir, Rebecca G. Wells, Orith Waisbourd-Zinman.

**Data curation:** Miri Dotan, Sophia Fried, Adi Har-Zahav, Orith Waisbourd-Zinman.

**Formal analysis:** Miri Dotan, Sophia Fried, Adi Har-Zahav, Orith Waisbourd-Zinman.

**Funding acquisition:** Orith Waisbourd-Zinman.

**Investigation:** Orith Waisbourd-Zinman.

**Methodology:** Orith Waisbourd-Zinman.

**Supervision:** Raanan Shamir, Rebecca G. Wells, Orith Waisbourd-Zinman.

**Writing – original draft:** Miri Dotan.

**Writing – review & editing:** Sophia Fried, Adi Har-Zahav, Raanan Shamir, Rebecca G. Wells, Orith Waisbourd-Zinman.

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
