## [Decision Letter · Decision Letter 0]

3 Jan 2022

PONE-D-21-36274Periductal bile acid exposure causes cholangiocyte injury and fibrosisPLOS ONE

Dear Dr. Waisbourd-Zinman,

Thank you for submitting your manuscript to PLOS ONE. After careful consideration, we feel that it has merit but does not fully meet PLOS ONE’s publication criteria as it currently stands. Therefore, we invite you to submit a revised version of the manuscript that addresses the points raised during the review process.  Please note the reviewers strong concerns that the data from the mouse model may not translate easily to the human disease in question. Please very carefully address the reviewers concerns

We look forward to receiving your revised manuscript.

Kind regards,

Aftab A. Ansari, PhD

Academic Editor

PLOS ONE

Journal Requirements:

Reviewers' comments:

Reviewer's Responses to Questions

**Comments to the Author**

1. Is the manuscript technically sound, and do the data support the conclusions?

Reviewer #1: Partly

2. Has the statistical analysis been performed appropriately and rigorously? 

Reviewer #1: N/A

3. Have the authors made all data underlying the findings in their manuscript fully available?

Reviewer #1: Yes

4. Is the manuscript presented in an intelligible fashion and written in standard English?

Reviewer #1: Yes

5. Review Comments to the Author

Reviewer #1: First and foremost, two preliminary remarks should be made here: biliary atresia (BA) is an inflammatory process of the intra- and extra-hepatic biliary tree, leading to ongoing fibrosis and deteriorating liver function. However, according to diagnostic patterns and and surgical treatment, BA ist only defined by the pathological findings along the extrahepatic bile ducts. This contradiction is frequently overseen, when etiologically directed research is performed. The second point is that there is so far no BA-animal model available, which displays precisely the pathomechanism and the course of the human disease. Taking this consideration into account, one should act cautiously when findings in mouse models or cell cultures shall be translationally interpreted.

The authors of the submitted study show that bile acid exposure to isolated extra hepatic bile ducts, which were harvested from adult and neonatal Bab/c-mice, results in cholangiocyte injury and fibrosis. They also demonstrate that ursodeoxycholic acid has a protective effect and reduces the destruction of the epithelial layer of the bile ducts. Limitations of the study are listed by the authors themselves, of which the argument concerning the exposure of the specimens to the bile salts was not restricted to the epithelium but to the bile duct in toto, is the most relevant.

Although the study provides interesting results about the toxic effect of bile salts to bile ducts and the already demonstrated protective value of ursodeoxycholic acid, it is not acceptable to discuss these findings in the context of the pathomechanism of BA. Hence, the observations of this interesting study should be imbedded in any other context while the relevance of the observations could be improved by showing that no isolated intra-hepatic bile ducts react the same way.

6. PLOS authors have the option to publish the peer review history of their article (what does this mean?). If published, this will include your full peer review and any attached files.

Reviewer #1: No

---

## [Author Response · Author response to Decision Letter 0]

21 Feb 2022

Response to reviewers:

Reviewer 1:

1. First and foremost, two preliminary remarks should be made here: biliary atresia (BA) is an inflammatory process of the intra- and extrahepatic biliary tree, leading to ongoing fibrosis and deteriorating liver function. However, according to diagnostic patterns and surgical treatment, BA is only defined by the pathological findings along the extrahepatic bile ducts. This contradiction is frequently overseen, when etiologically directed research is performed. 

Response: We thank the reviewer for this comment and agree that BA is a pan-cholangiopathy. At the time of BA diagnosis the extra-hepatic biliary tree is more effected compared to the intra-hepatic bile ducts; we thus hypothesize that the extra-hepatic cholangiocytes are either more sensitive to injury in the neonate or that there are hepatic mechanism compensating in part for the intra-hepatic cholangiocytes. Here in this study, we specifically aimed to determine the effect of increased permeability of the extra-hepatic bile ducts both to cholangiocytes and to the peri-ductal area. Based on the reviewer suggestion to modify the paper to reflect cholangiopathies in general and not specifically BA, we have edited the text accordingly. 

2. The second point is that there is so far no BA-animal model available, which displays precisely the pathomechanism and the course of the human disease. Taking this consideration into account, one should act cautiously when findings in mouse models or cell cultures shall be translationally interpreted.

Response: We thank the reviewer for this comment. We agree with this limitation of the study. Indeed mice research should be interpreted cautiously with future translational work to support those findings both from our labs and other groups. We have emphasized this well taken point in the discussion.

3. The authors of the submitted study show that bile acid exposure to isolated extra hepatic bile ducts, which were harvested from adult and neonatal Balb/c-mice, results in cholangiocyte injury and fibrosis. They also demonstrate that ursodeoxycholic acid has a protective effect and reduces the destruction of the epithelial layer of the bile ducts. Limitations of the study are listed by the authors themselves, of which the argument concerning the exposure of the specimens to the bile salts was not restricted to the epithelium but to the bile duct in toto, is the most relevant. The relevance of the observations could be improved by showing that no isolated intra-hepatic bile ducts react the same way.

Response: This is indeed an important point. We are not aware of a methodology that isolates intra-hepatic bile ducts as a whole rather than single cholangiocytes, and are not caplbe of performing it in our facility. Our study specifically aimed to determine the effect of bile acids exposure to the extra hepatic peri-ductal area and cholangiocytes in the basolateral side. We agree that the results do no imply on intra-hepatic bile ducts and we have refied our discussion to include that. 

4. Although the study provides interesting results about the toxic effect of bile salts to bile ducts and the already demonstrated protective value of ursodeoxycholic acid, it is not acceptable to discuss these findings in the context of the pathomechanism of BA. Hence, the observations of this interesting study should be imbedded in any other context. Response: We thank the reviewer for this comment. We have accepted this comment in full and substantially reduced the context of biliary atresia throughout the manuscript and adjugsted the discussed the findings in the context of cholangiopathies in general.

---

## [Editor Report · Decision Letter 1]

2 Mar 2022

Periductal bile acid exposure causes cholangiocyte injury and fibrosis

PONE-D-21-36274R1

Dear Dr. Waisbourd-Zinman,

We’re pleased to inform you that your manuscript has been judged scientifically suitable for publication and will be formally accepted for publication once it meets all outstanding technical requirements.

Kind regards,

Aftab A. Ansari, PhD

Academic Editor

PLOS ONE
---

## [Editor Report · Acceptance letter]

7 Mar 2022

PONE-D-21-36274R1 

Periductal bile acid exposure causes cholangiocyte injury and fibrosis 

Dear Dr. Waisbourd-Zinman:

I'm pleased to inform you that your manuscript has been deemed suitable for publication in PLOS ONE. Congratulations! Your manuscript is now with our production department. 

Kind regards, 

on behalf of

Dr. Aftab A. Ansari 

Academic Editor

PLOS ONE